# Intraoperative Cochlear Nerve Monitoring in Cochlear Implantation after Vestibular Schwannoma Resection

**Valerio Maria Di Pasquale Fiasca and Giulia Tealdo ***

Section of Otolaryngology, Otolaryngology Unit, Department of Neurosciences, University of Padova,
Via Giustiniani, 2, 35128 Padua, Italy
* Correspondence: giulia.tealdo@aopd.veneto.it; Tel.: +39-3480017494

**Abstract:** Background: The use of a cochlear implant (CI) for hearing rehabilitation after vestibular schwannoma (VS) resection is widely spreading. The procedure is usually performed simultaneously to tumor resection with a translabyrinthine approach. To ensure the best device function, assessing the integrity of the cochlear nerve is of primary importance. Methods: A narrative review of the literature on the present topic was carried out up to June 2022. Finally, nine studies were considered. Results: Electrically evoked auditory brainstem responses (eABR) is the most widely used method of intraoperative monitoring of cochlear nerve (CN) during VS resection, although its limits are known. It can be assessed through the CI electrode array or through an intracochlear test electrode (ITE). Variations of the graph are evaluated during the surgical procedure, in particular the wave V amplitude and latency. As tumor dissection progresses, the parameters may change, informing of the CN status, and the surgical procedure may be modulated. Conclusion: An eABR positive result seems to be reliably correlated with a good CI outcome in those cases in which a clear wave V is recorded before and after tumor removal. On the contrary, in those cases in which the eABR is lost or altered during the surgical procedure, the positioning of a CI is still debatable.

**Keywords:** vestibular schwannoma; cochlear implant; intraoperative monitoring

## 1. Introduction

General information—Vestibular schwannoma (VS), or acoustic neuroma as in a former definition, is a benign Schwann cell-derived tumor. This kind of neoformation is mostly sporadic (95%), single and unilateral, even if it is possible to find multiple and bilateral VS in patients affected by specific genetic syndromes such as neurofibromatosis type 2 (NF2, 5%). It originates from the layer of glial cells that envelopes the VIII cranial nerve. The VS most frequently develops from cells surrounding the vestibular branch of the nerve. It occupies the internal acoustic canal and then the cerebellopontine angle. Indeed, it comprises more than 80% of tumors of this skull base angle. During its progression, it compresses the structures that lie inside the canal [1,2]. VS can cause a variety of symptoms. Among them, hearing loss (often presenting as sudden and in high frequencies) and tinnitus are the most common. Other possible clinical presentations include dizziness, facial palsy, and imbalance. In most severe cases, the VS can compress the brainstem and cause hydrocephalus [2,3].

Available treatments—Today, there are different types of treatment available for the management of VS: observation, radiotherapy and surgery. Possible management choices depend on the characteristics of the tumor and the functional features of the patient related to the internal ear canal presented at the time of medical evaluation. For those cases in which surgical management is proposed to the patient, one of the main objectives is to preserve the functions of the internal ear canal structures, in particular the activity of the facial muscles (managed by the facial nerve) and the hearing sense. On the other hand, it is often unlikely to preserve the function of the vestibular organ. Surgical approaches

are usually divided into hearing preserving surgery (HPS) and non-hearing preserving surgery. HPS grants the possibility of dissecting the tumor from the nerve envelope within the cerebellopontine angle without damaging internal ear structures. Therefore, it provides the possibility to preserve natural preoperative hearing. In those cases where it is not possible to preserve the function of the internal ear canal structure, it is possible to try rehabilitation. The translabyrinthine approach is one of the most frequently performed [4–6]. This surgical approach allows the surgeon to reach the internal acoustic canal, through mastoidectomy and labyrinthectomy. Although this procedure provides less risk of postoperative neurosurgical complications and better surgical access for tumor dissection within the internal acoustic canal and internal ear structures, it results in profound hearing loss. This effect is due to the opening of the internal ear membranous labyrinth and the loss of perilymph. Therefore, the translabyrinthine approach is considered a treatment choice when hearing preservation is not likely to be successful, and HPS is excluded, as in the case of preoperative ipsilateral deafness or non-serviceable hearing [7,8].

Hearing function after surgery—One of the most important objectives in VS surgical treatment is preserving useful hearing. Although a translabyrinthine VS resection results in profound sensorineural hearing loss, it is possible to rehabilitate the patient by performing simultaneous or sequential positioning of a cochlear implant (CI). CI is a hearing device capable of directly stimulating the VIII nerve in the spiral ganglion through an array of electrodes surgically placed inside the cochlea [9]. The preservation of the CN is essential to allow this kind of rehabilitation. The surgeon should carefully preserve the integrity of the CN during surgery. Nowadays, this device is considered the best hearing rehabilitation choice for VS patients undergoing non-hearing-preserving surgery or in cases of failed HPS [10–13]. Unlike other hearing rehabilitation devices (such as bone-anchored and contralateral-routing-of-signal hearing aids), CI provides improvement in various hearing characteristics, such as sound localization and speech understanding. Nevertheless, VS patients rehabilitated with CI could not achieve the same speech outcomes levels described in the case of other aetiologies of hearing loss, because of the effect of the tumor on the CN or the internal ear structures [14]. Positioning of an IC can be performed simultaneously at the same surgical time as VS resection, or sequentially at a second surgical time. Arriaga and Marks published the first simultaneous vestibular schwannoma resection and cochlear implantation in 1995 [15]. This kind of procedure is more and more frequently performed. It grants some advantages. It allows to avoid a second surgery, reducing the risks related to the surgical acts. Moreover, the placement of the CI reduces the possibility of postoperative complications such as cochlear ossification or degeneration of spiral ganglion cells [16], which could affect the inner ear or the CN, leading to a worsening in the rehabilitation outcomes of CI. As reported by West et al. [17], the results of CI after VS surgery are very heterogeneous. There is no general agreement on the preoperative prognostic factors that influence the functional outcomes of postoperative IC after treatment for this kind of retrocochlear disease. It is acknowledged that the anatomical and functional integrity of CN after tumor resection is of utmost importance to achieve useful postoperative CI function. During resection of the tumor from the Schwann cells, as previously stated, the surgeon must macroscopically manage to avoid damage to the CN. On the other hand, the precise assessment of the functional integrity of the CN in the preoperative and intraoperative setting is one of the most challenging tasks in both the simultaneous and the two-stage sequential surgical procedure. Macroscopic preservation of the neural structure could not be sufficient to ensure postoperative functional hearing. CN evoked compound action potentials (eCAP) and eABR are tests which have been used as intraoperative monitoring tools to evaluate nerve integrity. For two-stage procedures, promontory stimulation can be used to assess nerve conduction [18,19]. The reliability of all these methods to investigate the CN status and predict postsurgical CI outcomes, and whether hearing function would be restored after surgery in the event of implantation, is the object of research. The number of studies focused on this topic remains limited in literature and it seems difficult to find a general agreement.

Aim—The aim of this narrative review is to describe the current knowledge in the methods and results of CN monitoring. The study focused on intraoperative procedures, considering patients who underwent simultaneous CI positioning after vestibular schwannoma resection.

## 2. Materials and Methods

For this narrative review, we considered the most relevant studies published in three different databases, PubMed, Embase and Scopus, up to June 2022. The literature search was performed by the two authors independently The words "vestibular schwannoma", "cochlear implant" and "intraoperative monitoring" were entered in different combinations. All the retrieved publications were evaluated to identify the most relevant ones. Duplications or aggregations of preexisting data were excluded. The reference lists of selected articles were also analyzed to identify additional studies. Every misalignment the authors had regarding article eligibility was resolved through discussion. All the retrieved publications were evaluated to identify the most relevant ones, and nine articles were identified.

## 3. Results

The most relevant studies on intraoperative CN monitoring in patients who underwent CI after vestibular schwannoma resection and their results are shown in Tables 1 and 2.

**Table 1.** Intraoperative monitoring of CN in cochlear implantation after vestibular schwannoma resection.

| First Author, Publish Year | Study Design | Number of Patients | IOM Type | Intraoperative eABR before Resection (Present:Absent) | Intraoperative eABR after Resection (Present:Absent) | Cochlear Implantation |
|---|---|---|---|---|---|---|
| Lloyd, 2014 [20] | Retrospective case series | 2 | eABR | 2:0 | NR | 2 |
| Lassaletta, 2017 [10] | Prospective case series | 10 | eABR with test electrode | 10:0 | 10:0 | 10 |
| Rahne, 2018 [21] | Case report | 1 | eABR with CI array | 1:0 | 1:0 | 1 |
| Kasbekar, 2019 [22] | Case report | 1 | eABR with test electrode | 1:0 | 0:1 | 1 |
| Patel, 2019 [23] | Case report | 1 | eABR with CI array | 1:0 | 1:0 | 1 |
| Dahm, 2020 [24] | Prospective case series | 5 | eABR with test electrode | 5:0 | 3:2 | 5 |
| Medina, 2020 [25] | Prospective multicenter case series | 21 | eABR with test electrode | 13:8 | 9:12 | 15 |
| Butler, 2021 [26] | Retrospective case series | 3 | eABR with test electrode | 3:0 | 1:2 | 3 |
| Weiss, 2021 [27] | Case report | 1 | eABR with CI array | 1:0 | 1:0 | 1 |

IOM: intraoperative monitoring; NR: not recorded; CI: cochlear implant; eABR: electrical-evoked auditory brainstem response.

**Table 2.** Postoperative findings after vestibular schwannoma resection and cochlear implantation.

| First Author, Publish Year | Cochlear Implantation | Post-Implantation eABR (Present:Absent) | CI Activation eABR (Present:Absent) | eABR Follow Up (Present:Absent) |
|---|---|---|---|---|
| Lloyd, 2014 [20] | 2 | 2:0 | 2:0 | 2:0 |
| Lassaletta, 2017 [10] | 10 | 10:0 | NR | NR |
| Rahne, 2018 [21] | 1 | NR | NR | NR |
| Kasbekar, 2019 [22] | 1 | 1:0 (after use of papaverine) | NR | NR |
| Patel, 2019 [23] | 1 | 1:0 | 0:1 | 0:1 |
| Dahm, 2020 [24] | 5 | NR | NR | NR |
| Medina, 2020 [25] | 15 | 10:5 | NR | NR |
| Butler, 2021 [26] | 3 | 3:0 | 2:1 | 1:2 |
| Weiss, 2021 [27] | 1 | 1:0 | NR | 1:0 |

IOM: intraoperative monitoring; NR: not recorded; CI: cochlear implant; eABR: electrical-evoked auditory brainstem response.

The nine studies included in this work dated from 2014 to 2021. In total, 45 patients were enrolled in the analyzed studies. Most of those works were in the form of case reports. In all studies, an eABR was registered before and after surgery, and again, when performed, after positioning of the CI. Intraoperative eABR was measured using an intracochlear test electrode (ITE) in five papers; the cochlear implant was used to assess eABR in four articles, and in one last work, this information is not reported. In 39 patients, a cochlear implantation followed VS surgical treatment. The decision to perform the cochlear implantation depended on the integrity of the CN after the surgical procedure: in some cases, where the surgeon noticed that the CN was damaged or interrupted, the CI position was avoided. The absence of eABR after tumor resection was not considered a criterion for avoiding cochlear implantation. Among the implanted patients, most showed a normal eABR after surgery. Unfortunately, many of the present studies do not show CI activation and follow-up eABR.

## 4. Discussion

### 4.1. Evoked Compound Action Potentials (eCAP)

Evoked compound action potentials (eCAP) are a useful tool used to assess the integrity of CN, even in the intraoperative setting. In this kind of test, neural responses arising from the peripheral part of the auditory nerve are collected and analyzed. These responses represent the activation of a population of electrically stimulated CN fibers. Electrical activity can be recorded directly on a surgically exposed nerve (as in animal samples) or from an intracochlear electrode such as a cochlear implant electrode array. Therefore, the eCAP can describe the activity of CN cells. eCAP may be useful, especially in CI fitting, to estimate subjective thresholds and maximum comfortable loudness levels [28,29]. It has also been used as a routine procedure to verify implant function and ensure responses of the auditory nerve to electrical stimulation during surgery in nontumor patients [30]. Evidence suggests that eCAP responses are able to reflect the health status of auditory nerve fibers near the recording electrodes. They could play an important role in determining the outcomes of CI. Additionally, eCAP can be assessed along with eABR, with the aim of achieving the best comprehension of the auditory pathway status [23].

### 4.2. Electrically Evoked Auditory Brainsteam Response (eABR)

Electrical ABR (eABR) is a bioelectric neural activity evoked in response to stimulation of the CN. It allows for assessing the integrity and the function of the CN as the first step of the auditory pathway. The eABR is obtained using an electrical stimulus that can be applied to different parts of the medium or internal ear, such as the promontory, the round window or the cochlea [31]. In recent years, the use of ABR has been recognized as a reliable method for evaluating the neural pathway that carries hearing information. Furthermore, many studies described the use of eABR as a useful tool to assess the residual function of the CN in patients affected by VS.

There are different possible positions to record the eABR. It can be tracked with an electrode positioned on the promontory, an ITE, or by delivering the stimulation using a CI electrode. eABR potentials are small compared to background electrical brain activity. To allow the recording of these tests and overcome other disturbing electrical signals, several thousand samples of the electrical stimulus and responses are collected. Then, these samples are averaged to create a single and distinct auditory evoked potential. The resulting graph describes the different steps of the auditory pathway. A normal eABR indicates normal electrical conduction along the pathway and therefore integrity in CN fibers [32]. Some disadvantages are known for this kind of evaluation, such as (i) the delay between surgical trauma on the nerve and recording of ABR waves, and (ii) false positives related to anaesthesia, hypothermia or irrigation [33]. The quality of the eABR waveform can be assessed with the scoring criteria published by Walton et al. [34] and compared. The Walton criteria for the shape of the eABR wave are shown in Table 3.

**Table 3.** Walton criteria [34] for the eABR waveform.

| Score | Wave II | Wave III | Wave V | Amplitude Wave V |
|---|---|---|---|---|
| 3 | Yes | Yes | Yes | >0.5 µV |
| 2 | Yes, reduced amplitude | Yes, reduced amplitude | Yes | <0.5 µV |
| 1 | No | No | Yes | <0.5 µV |
| 0 | No | No | No | 0 |

In the eABR graph, the most interesting parts to assess the integrity and function of the CN are those of the III and V waves. These waves represent the signal reaching the olivary complex and the lateral lemniscus. Because of their robustness, wave III and in particular the wave V are considered to reflect the CN integrity. On the other hand, I and II are known to be unstable and affected by artefacts caused by intracochlear electrical stimulation [35,36]. Other features of the eABR graph are usually assessed, such as latencies of single waves I, III and V and interwave latencies of waves I–III, I–V and III–V. The presence of wave V during and after the VS resection, more than the other parts of the eABR graphs, has shown a prognostic power to predict hearing preservation. Instead, in the case of eABR alteration, the degree of postoperative deficits can be difficult to predict [33]. A 50% decrease in wave amplitude or a 1 ms increase in peak latency for wave V are usually considered signs of CN damage. The complete loss of wave V has been described as a factor correlated with hearing loss and a poor functional outcome of CI [37].

4.2.1. eABR via CI Array

The most used method to perform an eABR assessment during VS resection is the recording via the positioning of the CI electrode array into the cochlea and the delivery of electrical stimulation. The presence of a wave V has been reported to be a reliable index to confirm the positioning of CI [21,23,27]. In contrast, in the event of negative results, eABR is still considered not sufficient to deny the positioning of the device. Most studies on this type of monitoring are case reports or based on a very limited number of patients.

Patel et al. [23] described the case of a neuromonitoring performed using a CI electrode array (Advanced Bionics® HiRes Ultra 3D CI with a HiFocus SlimJ electrode (Advanced Bionics LLC, Valencia, CA, USA)) in a patient affected by a sporadic VS with asymmetric hearing loss. The intracochlear electrodes were used to record CN acoustic potentials (CNAP) through continuous neural response imaging (NRI) and eABR to assess the integrity of the cochlear during the translabyrinthine craniotomy for tumor resection. The authors stated that assessing CNAP reduces the delay between surgical maneuvers and neural reaction and described a correlation between pressure or traction on the CN and changes in recordings. This tool provides useful information on potentially traumatic manipulation of the CN. In the case of CNAP activation, an evaluation of an updated eABR waveform was performed. The hearing thresholds measured during the CI follow-up were higher than the intraoperative CNAP thresholds. Fluctuations in eABR wave V were measured during tumor resection, but it showed complete recovery after surgery. The eABR showed an absence of a V wave during the CI activation assessment, and a lack of sound detection was reported.

In 2021, Weiss et al. [27] described a case of intraoperative CN monitoring, evaluating the eABR with a CI. The implant (Cochlear Nucleus CI622® (Cochlear Intl, Sydney, Australia)) was placed after a microsurgical resection of an intrameatal VS using a translabyrinthine approach. The eABR showed constantly reproducible waves III and V with stable amplitudes and latencies, which could point to a preservation of the CN. One month after CI activation, the patient reached useful hearing. He was able to receive and answer telephone calls, and after three months his monosyllable recognition was 70% at 65 dB.

Rahne et al. [21] reported a case of a patient with NF2 with intralabyrinthine schwannoma. After tumor resection and cochlear implantation, they performed an eABR using the positioned device, generating positive responses. Prolonged latencies of the eIII and

eV waves and normal interpeak latencies were recorded. eCAPs were assessed to collect information regarding the peripheral CN. Eventually, the patient showed good early audiological outcomes (one month later the open-set word recognition was 100% monosyllables).

Lloyd et al. have reported the cases of two NF2 patients who underwent simultaneous translabyrinthine VS resection and CI positioning [20]. The integrity of the CN was monitored with a "Cueva electrode": a "golf club" electrode placed in the circular window niche [38]. eABR and CNAP were recorded, confirming a positive CN action potential. Simultaneous nucleus freedom CI were inserted. A benefit was reported from the device and regular wear of the cochlear implant in both cases, even in the presence of contralateral hearing. The authors stated that a robust eABR is likely to be associated with a good CI function. On the contrary, if the eABR is absent, they suggest a hybrid implantation of ABI and CI.

Butler et al. [26] published the case of three NF2 patients who underwent translabyrinthine VS resection. During the procedure, an intraoperative monitoring of the CN was performed with an eABR. At the end of the surgery, a CI was positioned. Two were tested by CI (MED-EL[®] (Innsbruck, Austria) Synchrony Flex 28); the third patient first received an ITE (Acoustic Nerve Test System). All patients showed good eABR, but different CI outcomes. The first had a good CI function at 9 months of follow-up, but the function started to decline at 1 year and ended with a complete absence of activity. The second patient reached fluctuating audibility and limited speech understanding at 3 months postoperatively. The third patient reported good sound quality, word comprehension and sound localization, with daily use of the device.

### 4.2.2. eABR through ITE

Intraoperative neuromonitoring of CN integrity during VS resection recording eABR with an ITE manufactured by MED-EL[®] (Innsbruck, Austria) has been reported in various publications. ITE can assess CN function, registering the eABR while surgery is performed near the CN [10,22,24,25]. The ITE consists of four electrode contacts: three of the electrodes are placed directly into the cochlea and the fourth reference electrode is placed under the temporal muscle. The insertion should be careful, fixing it in the round window to prevent movement during surgery. Moreover, it allows us to evaluate CN activity before placing a CI electrode array. The use of the ITE could allow for not wasting a CI in case of irreversible nerve damage at the end of surgery. Moreover, it has been described as a viable alternative to electrode array monitoring.

In 2017, Lassaletta et al. [10] analyzed postoperative eABR in a group of 10 patients with VS, both with an array of ITE and CI electrodes. They demonstrated the presence of similar outcomes (latencies and amplitudes of waves III and V) between the two methods of recording, without significant differences.

Medina et al. [25] published a study in 2020 aiming to assess the usefulness of eABR obtained intraoperatively with the ITE after resection of a VS. The work also aimed to calculate this eABR diagnostic accuracy to assess the functionality of the CN for CI. eABR was performed in monopolar mode before labyrinthectomy, after tumor resection and after CI insertion using the CI electrode array (electrodes 5 and 7). The eABR was evaluated according to the Walton classification [27]. The study incorporated 21 patients with different hearing characteristics, of whom just 15 received CI. The decision to not place CI in the remaining six patients was motivated by the CN section during the surgical act (three cases) or the finding of a traumatized CN with the absence of EABR after tumor exeresis (three cases). One of the not-implanted patients received an auditory brainstem implant. Among implanted patients, nine cases had positive eABR and had auditory perception with CI. In the remaining five cases, a negative eABR was measured and four did not reach auditory perception with the implanted device. These findings confirmed the usefulness of ITE in predicting CN integrity after tumor removal, with an accuracy of 93%.

Dahm et al. [24] described a series of five patients affected by sporadic VS, in which a simultaneous resection was performed via a translabyrinthine approach and cochlear implantation. Intraoperatively, eABR was used to assess nerve integrity and the results

were correlated with postoperative hearing. MED-EL® ITE was applied to perform an eABR and as a monitoring tool when performing surgery close to the CN. The author described some difficulties, such as signal delay in monitoring and the presence of recording artefacts. All patients received CI. A clear wave V could be identified in three patients before and after tumor removal with ITE. Since surgeons evaluated the CN intact in each patient after surgical procedures, CI was placed in all of them. Patients underwent a 12-month follow-up and were evaluated for hearing function and sound localization. Among patients with clearly identified V wave on the intraoperative eABR test, two reached 50% speech comprehension at 61.9 and 65.0 dB and improved sound localizations, while another achieved sound perception but no speech understanding or sound localizations. The two patients in which no V waves were identified had no sound perception with the CI at activation. All three patients with sound perception are daily users and very satisfied with the CI performance.

### 4.3. Promontory Stimulation

Promontory stimulation is used in sequential cochlear implantation after VS resection (for example, in case of hearing preservation surgery failures). It can confirm CN fiber survival and can predict useful speech benefits after implantation [39]. Negative results, on the other hand, do not exclude patients from using a CI successfully [40]. Promontory stimulation was one of the first tests used to assess the residual function of CN prior to cochlear implantation. It is performed by positioning an extracochlear trans tympanic needle electrode on the promontory and sending a stimulation in the form of electrical pulses. The amplitude of the pulses increases until a response is recorded [20]. Promontory stimulation can represent a reliable estimate of the CN status after surgery, although in some cases responses may not be detected, due to artefacts or other causes [41]. Arnoldner et al. [42] proposed a new scoring system to predict the CI outcome and therefore CN preservation during VS surgery. They retrospectively evaluated the results of cochlear implantation after VS resection and correlated them with some clinical-pathological parameters, such as (i) tumor size, (ii) presence of preoperative eABR (assessed with promontory stimulation), (iii) preoperative hearing and (iv) relationship of VS with modiolus. Therefore, the patients were divided into four classes with different probabilities of favorable CI results: (I) class I had a very high chance of positive CI results after tumor resection, (II) class II had a worse chance of favorable CI results and (III) class III and IV had negative CI results.

### 5. Conclusions

Nowadays, the preservation of useful hearing after VS surgery is considered one of the most important surgical outcomes. Among the possible surgical options, the translabyrinthine surgical approach is one of the most frequently performed for VS treatment. This kind of procedure provides many advantages in terms of access to the cerebello-pontine angle and tumor management, but results in ipsilateral sensorineural profound hearing loss. Patients who undergo this kind of approach can be rehabilitated with CI. This device can restore a useful hearing similar to natural preoperative hearing. The positioning of the implanted device can be performed simultaneously with the VS resection. The CI is a device capable of directly stimulating the CN. Assessing the integrity of the nerve and the entire auditory pathway is of primary importance for hearing rehabilitation after VS resection. Only a preserved CN can be stimulated by the CI and provide the best functional results. The eABR usefulness for evaluation of CN activity and integrity is being studied. This test is feasible during translabyrinthine VS resection. This kind of assessment can be performed in both perioperative and intraoperative settings, and many strategies have been tested to fulfil the task. Some changes in the eABR graph are expected during the surgical procedure. Interpretation of the eABR signal starts with a wave V measurement of amplitude and latency. The wave III and the relationship between these two different waves can provide interesting information regarding CN status. As tumor dissection progresses the eABR parameters may change, informing about the status of the CN and possibly reflecting

the effect of the surgical procedure on the nerve. Consequently, the surgical procedure can be modulated. After tumor dissection, an intact or well-preserved wave V suggests that the CN has not been damaged during the surgical procedure. Unfortunately, the intraoperative e-ABR may not be strictly correlated with the postoperative CI outcome. Although clear wave V before and after tumor removal recorded with eABR seems to correlate reliably with a good result, in those cases in which eABR is lost during the surgical procedure, positioning of a CI is still debatable. Some patients reached a speech perception despite the absence of intraoperative eABR. The results of this monitoring are promising, but its reliability to predict the CI needs further testing. More studies with homogeneous series and a higher number of patients are necessary.

**Author Contributions:** Conceptualization, V.M.D.P.F. and G.T.; methodology, G.T.; software, V.M.D.P.F.; validation, G.T.; formal analysis, G.T.; investigation, V.M.D.P.F. and G.T.; data curation, V.M.D.P.F.; writing—original draft preparation, V.M.D.P.F.; writing—review and editing, V.M.D.P.F. and G.T.; visualization, G.T.; supervision, G.T.; project administration, G.T. All authors have read and agreed to the published version of the manuscript.

**Funding:** This research received no external funding.

**Institutional Review Board Statement:** Not applicable.

**Informed Consent Statement:** Not applicable.

**Data Availability Statement:** No new data were created or analyzed in this study. Data sharing is not applicable to this article.

**Conflicts of Interest:** The authors declare no conflict of interest.

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
