# Peer review of "Intraoperative Cochlear Nerve Monitoring in Cochlear Implantation after Vestibular Schwannoma Resection"

_audiolres, doi:10.3390/audiolres13030035_

Round 1

Reviewer 1 Report

This narrative review paper provides readers of this journal with useful information regarding cochlear implantation after vestibular schwannoma resection. This topic will gain more attention as hearing rehabilitation will become more important in the management of vestibular schwannoma. This reviewer appreciates the valuable work of the authors. Please check some errors in spelling and grammar.

Round 2

Reviewer 1 Report

This paper is acceptable in the revised version.

Author Response

Dear reviewer,

Thanks for your comments.

Dr. G. Tealdo and V. M. Di Pasquale Fiasca

Reviewer 2 Report

The introduction has been expanded in this revised version of the manuscript, please consider subheadings to improve readability.

Most of the conclusion regards eABR, is this the gold standard in your opinion? I see contradictions between lines 333-335 and 335-337.

According to SANRA (scale for the quality assessment of narrative review Articles) a convincing narrative review presence evidence for key arguments. Considering add more mentions of study designs and level of evidence alongside the manuscript.

Some minor typo: Line 68 "BAHD": missing definition for this acronym. Line 156 "IC" did you mean "CI"? line 302: "Arnone et al." is correct?

Author Response

Dear reviewer,

As you can see in the new version of our article, we followed the suggestions of adding subheadings in the introduction, which we expanded, as you noticed. We hope you find our introduction better organised. We made corrections regarding the conclusion lines, which you correctly commented on how the text could have been misleading.
We tried to emphasise the advantages of eABR in intraoperative cochlear nerve monitoring during vestibular schwannoma, but also we underlined how currently this test still has no direct correlation with the postoperative outcomes of cochlear implant.
We found your suggestion about the importance of providing more information on the mentioned studies very interesting. Therefore, we decided to add a column in Table 1, where we cited all the designs of the mentioned studies. As you will see, the number of patients considered is relatively small, and most studies are conducted in the form of case reports or limited case series. We think this depends on the specificity of the topic.

Also, we change "BAHD" into “bone-anchored and contralateral-routing-of-signal hearing aids“. And update CI in line 156, change Arnoldner et al in line 304.

We hope you will find our new version interesting.
Thanks for your attention,
Dr. G. Tealdo and V. M. Di Pasquale Fiasca